# Study of Rock Crack Extension under Liquid Nitrogen Low-Temperature Fracturing

**Chunyan Bao \*, Meng Zhou and Yuexiang Cai**

Rock Mechanics and Geological Hazards Experimental Center, School of Civil Engineering, Shao Xing University, Shaoxing 312000, China; a1336286391@163.com (M.Z.); a809965147@163.com (Y.C.)
\* Correspondence: fengniaobcy@163.com

**Abstract:** Shale gas is a promising new energy source stored in shale. This research aims to study the laws of rock crack initiation and propagation under the low-temperature fracturing of liquid nitrogen, explore the influencing factors of the shale reservoir fracturing effect, and identify the method that achieves the best fracturing effect and obtains the highest economic benefits. Herein, a visualized physical experiment of the liquid nitrogen effect is carried out, and the fracture process of a numerical model under cold shock is simulated to analyze the effect of homogeneity on shale crack propagation. The results show that two different crack development modes could be observed in the field test. The first one was the horizontal plane radial crack caused by longitudinal thermal shrinkage, and the other one was the vertical tensile crack caused by circumferential shrinkage. A certain interval length was frequently found between the horizontal cracks. The crack propagation of the specimens with different homogenization degrees was basically distributed in the direction perpendicular to the liquid nitrogen contact surface. When the homogenization degrees were m = 2 and 5, the crack surface was rough and the microfracture was disordered and dotted around the crack tip. When m ≥ 10, the dotted damage around the crack tip did not appear, and the crack propagation was close to the results obtained from using the homogeneous materials. Finally, this work simulates the fracture process of a circular hole plane model under cold shock, analyzes the influences of heat transfer coefficient, in situ stress and other parameters on shale temperature, minimum principal stress distribution, and crack propagation, and discusses the optimal method to improve the heat transfer coefficient. The results show that increasing the heat transfer coefficient can increase the tensile stress value and influence the range of the contact boundary, making the rock more prone to cracking and resulting in greater crack development and a better crack initiation effect. The lateral stress coefficient affects the propagation direction of the cracks, and the propagation depths of low-temperature cracks were found to be deeper in the direction of larger principal stress.

**Keywords:** shale; shale gas; liquid nitrogen; RFPA; crack propagation

## 1. Introduction

Shale gas, a clean and high-quality unconventional natural gas energy, is abundant in global reserves. Although the combination of horizontal wells and multistage hydraulic fracturing has become the key technology for the effective development of shale gas resources [1,2], it has greatly changed the expected production of unconventional energy. However, differences still persist between mining potential and mining capacity (mainly in terms of technical recoverable reserves). This situation is mainly because of the special and complex shale gas reservoir environment in China and the series of problems caused by the use of hydraulic fracturing, such as formation damage, water shortage, reflux difficulty, and environmental pollution [3,4], which all limit the development of shale gas. The use of low-temperature fluid injection fracturing technology, such as liquid nitrogen, is one of the ideas to solve this problem.

Unlike the commonly used high-pressure hydraulic fracturing, Li Lianchong et al. [5] carried out the hydraulic fracturing of shale on the basis of the basic equations of classical seepage mechanics, and concluded that the hydraulic process is a typical damage process of coupled seepage and stress. Zhang Liaoyuan et al. [6] established a numerical calculation model of hydraulic fracturing in bare vertical wells under large-scale three-dimensional stress conditions based on the RFPA3D program of the parallel finite element method, observed the effect of the seepage field distribution of the hydraulic fracturing model and the change of the minimum principal stress, analyzed the acoustic emission law of hydraulic fracture extension, and summarized the mechanisms of fracture initiation, extension, and the elongation of the hydraulic fractures. Liquid nitrogen low-temperature fracturing is the combined effect of four positive factors, including liquid nitrogen pressure, heat-induced stress, rock embrittlement, and nitrogen expansion [4]. Thermal stress caused by the temperature difference between rock and injected cold fluid plays an important role in crack initiation and propagation [7]. At present, some basic research on liquid nitrogen low-temperature fracturing has been carried out. Tang Shibin et al. [8] were the first to apply thermoelastic theory to obtain theoretical solutions for the evolution of internal temperature and stress in a flat plate model subjected to cold shock on one side, apply numerical simulation methods to study the crack expansion process under the effect of rock cooling shock, and analyze the effect of the heat transfer coefficient on rock temperature and minimum principal stress distribution. There are also some fracturing studies that use liquid nitrogen to produce a low-temperature effect, e.g., Grundmann et al. [9] pumped liquid nitrogen into Devonian shale wells at conventional flow rates and pressures. The results showed that the initial production of the well increased by 8%, and thermal cracks orthogonal to the main crack surface were expected to occur. Cai Chengzheng et al. [10–15] evaluated the rock damage and crack effect caused by the low-temperature cooling of liquid nitrogen via scanning electron microscopy, nuclear magnetic resonance, the acoustic emission (AE) test, and uniaxial compression. The damage degree was affected by lithology, porosity, and water saturation. Cha et al. [16,17] used an experimental device that they designed to study the permeability changes in rock treated with liquid nitrogen at a low temperature under true triaxial confining stress and the characteristics of microcrack propagation that it displayed. The above-mentioned experiments show that the injection of liquid nitrogen will generate new cracks and promote the propagation of existing cracks, increasing the fracturing effect inside the rock. Therefore, understanding the mechanism of this phenomenon and the law of crack propagation in low-temperature fracturing is of great significance for the field application of low-temperature fracturing.

However, crack formation and propagation under liquid nitrogen cold impact on shale are regarded as fast and highly complex processes, and the real-time acquisition of data related to the liquid nitrogen low-temperature fracturing process is difficult to control. Zhang et al. [18] conducted a numerical simulation of this phenomenon to obtain real-time data of the temperature field and stress field in the process of rock fracturing at a low temperature with liquid nitrogen. Yang et al. [19] simulated and predicted the productivity of gas wells using this technology and analyzed the economic benefits of 10 years of production. Additionally, the subsurface conditions in which shale gas is located are very complex, and there will be discrepancies between realistic and experimental models [20]. Both Amin [21] and M. Soleimani [22] have proposed integrated strategy approaches which use well data, 3D seismic data, and regional geological information to perform numerical simulations. This notion indicates that numerical simulation can overcome adverse factors in laboratory conditions, and it is an effective method to understand the mechanism.

Therefore, with the shale gas reservoir as the research object, our study examines low-temperature fracturing technology combined with visualization, laboratory tests, and numerical simulation methods, uses the means of RFPA software and a high-speed camera, and analyzes the different homogeneous degrees and heat transfer coefficients of shale conditions, such as the internal stress field, temperature field, and the change rule of the law of crack extension. During the simulation of the low-temperature fracturing of shale

with liquid nitrogen under cold impact, the law of crack initiation and propagation is highlighted, the influence of sensitive factors such as the heat transfer coefficient and in situ stress on crack propagation are studied, the relationship between each factor and crack development mode is obtained, and how to improve fracturing effect and increase energy production is discussed.

The paper contains the following four sections: a visual physical experimental analysis of the liquid nitrogen effect; a feasibility study on low-temperature frozen shale using a numerical simulation method; the numerical simulation of liquid nitrogen low-temperature fracturing; and the effect of in situ stress on crack propagation.

## 2. Visual Physical Experimental Analysis of Liquid Nitrogen Effect

### 2.1. Sample Preparation

In this experiment, an HT55X acrylic resin molding material from Sumitomo Chemical Co., Ltd. was selected as the material for the transparent model. The film was pressed to make a cylindrical sample of Φ100 mm × 200 mm, and the borehole was simulated. The sample size is shown in Figure 1, wherein the circular point indicates the tip position of the thermocouple. The thermodynamic parameters of the common rock and acrylic are shown in Table 1, according to the performance table of the acrylic materials issued by the product company and the related research results of the thermodynamic parameters of rock at room temperature.

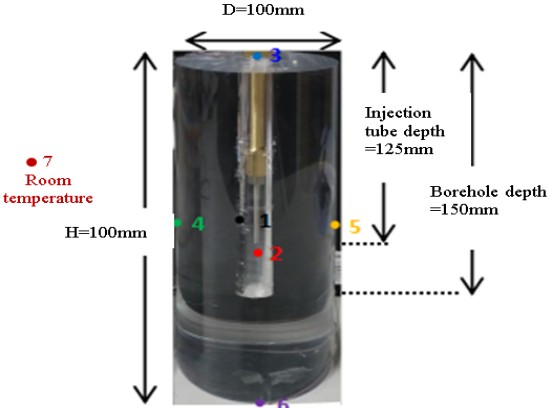

**Figure 1.** Schematic of the acrylic sample size (The dot indicates the position of the thermocouple).

**Table 1.** Mechanical parameters of the rock and acrylic.

| Parameter Name | | Medium | |
|---|---|---|---|
| | | Rock | Acrylic |
| Density (kg·m$^{-3}$) | | 2410 | 1168 |
| Mechanical property | Compressive strength (MPa) | 10–500 | 90–130 |
| | Tensile strength (MPa) | 2–40 | 50–77 |
| | Elastic modulus (GPa) | 5–80 | 3 |
| | Poisson's ratio | 0.05–0.4 | 0.37 |
| Thermal performance | Coefficient of thermal conductivity (W·m$^{-1}$·°C$^{-1}$) | 2.67 | 0.19 |
| | Thermal capacity (J·m$^{-3}$·°C$^{-1}$ × 10$^6$) | 1.9 | 1.8 |
| | Thermal expansion coefficient (°C$^{-1}$ × 10$^{-5}$) | 0.3 | 6–9 |
| Optical property | Transmittancy (%) | – | 91 |
| | Fuzzy coefficient (%) | – | 0.9 |
| | Refractive index (%) | – | 1.49 |

According to the above table, the deviation of the physical thermodynamic parameters between the two materials is small, and the internal structure is consistent. However, the resin materials showed obvious plastic deformation in the mechanical test. Wang Li [23], Wang

Xianggang [24], Lu Yongrong [25], and Lin Hengxing [26] modified the other transparent resin materials by using a low-temperature freezing method to change this situation. The brittleness of the resin materials can be significantly increased by using the low-temperature freezing method. In this study, the effect of low-temperature fluids, such as liquid nitrogen, on rock fractures (25 °C room temperature and −196 °C liquid nitrogen temperature) was highlighted, and we believed that this would promote the brittleness of the resin materials during the test.

### 2.2. Experimental Apparatus and Program

Figure 2 is a schematic of the liquid nitrogen low-pressure impact experiment. In this experiment, liquid nitrogen was extracted from the liquid nitrogen storage tank with a liquid nitrogen extraction device, and the pressure difference was obtained with a pedal cylinder. Meanwhile, the vent valve was closed, and the valve of the hose connection was opened. With the increase in pressure, liquid nitrogen was transported to the specimen through the insulating layer sleeve hose, injected into the borehole, and directed to the outlet. Figure 3 shows the flow path after the liquid nitrogen injection. The blue arrow denotes the liquid nitrogen that flows into the wellbore in the tube. The red arrow is the nitrogen that leaves the wellbore after the liquid nitrogen contact sample is evaporated. The figure demonstrates that liquid nitrogen makes contact with the sample at room temperature to absorb a large amount of heat, and gradually vaporizes into nitrogen. At the beginning of the test, the liquid nitrogen in the injection well was mainly nitrogen. The liquid–gas mixture was gradually transformed into the wellbore. Furthermore, epoxy resin was used as a sealing treatment between the sample and the metal device to ensure a smooth flow path. After injection, nitrogen flowed into the interface and opened the vent valve. Accordingly, nitrogen flowed into the outdoors from the exhaust hose to avoid indoor asphyxiation due to the discharge of large amounts of nitrogen.

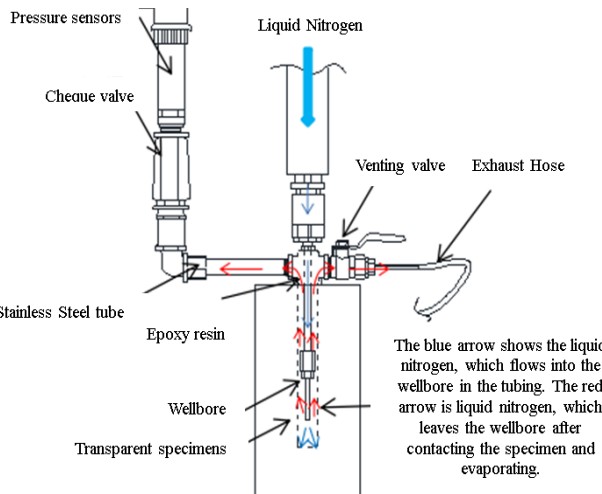

**Figure 2.** Schematic of the liquid nitrogen low-pressure impact experiment.

The wellbore contact surface was rapidly cooled through the rapid flow of liquid nitrogen in the wellbore, and the thermal gradient was maximized as far as possible. Liquid nitrogen was always maintained in a low-pressure environment of about 50 kPa throughout the flow process. The data acquisition system composed of a thermocouple, pressure sensor, and high-speed camera could rapidly respond to certain changes, such as temperature, and real-time monitor and record the borehole pressure, sample temperature, liquid nitrogen consumption, and crack propagation process.

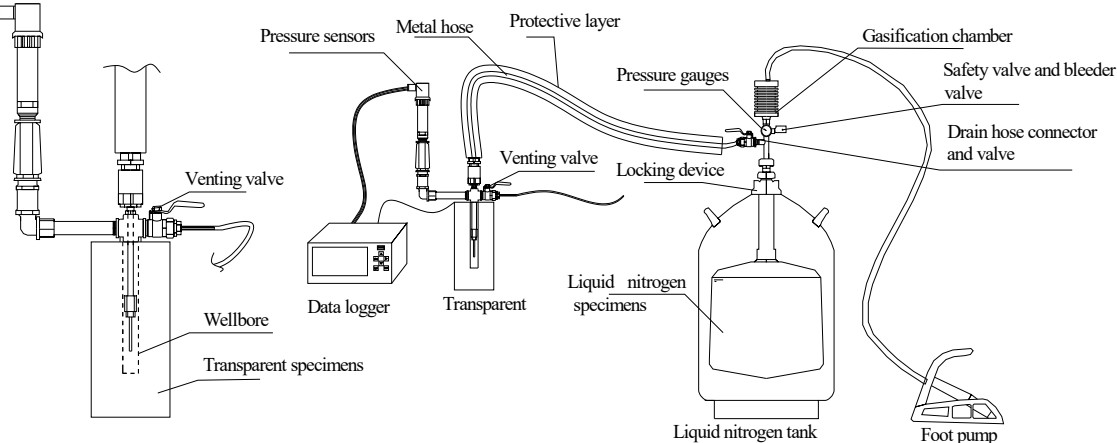

**Figure 3.** Low-pressure impact path of liquid nitrogen.

## 2.3. Temperature and Pressure

The total experimental duration of samples 1 and 2 was 25 min. During the whole release process (the first 9 min), the pressure in the hole was always in the range of 40–50 kPa (Figure 4). The drainage hose valve was fully opened, and no closure could be found in the middle.

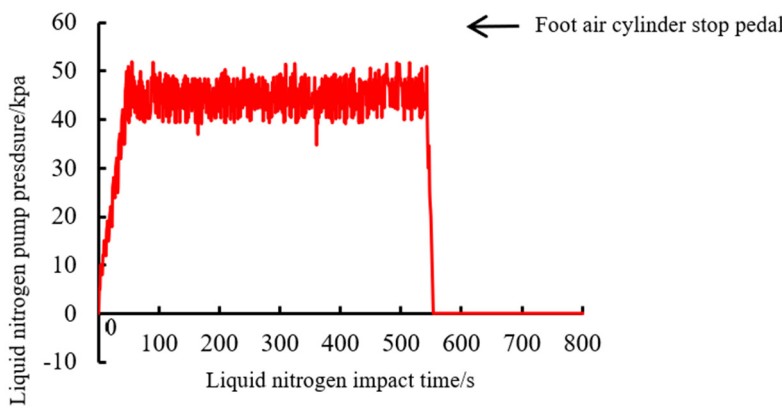

**Figure 4.** Pressure variation of the liquid nitrogen pump during the cryogenic thermal shock experiments.

Figure 5 shows the temperature variation of the thermocouple tip during the liquid nitrogen impact test. During the test process, the sample affected by the liquid nitrogen temperature rapidly decreased, reaching its boiling point within 3 min (−196 °C). After 9 min, the pedaling gas cylinder was stopped. The pressure of the liquid nitrogen pump on the liquid nitrogen storage tank rapidly decreased, and the inflow of liquid nitrogen immediately decreased until no nitrogen was discharged. In this process, the internal temperature of the sample borehole (such as TC1 and TC2 in Figure 1) rapidly rewarmed until it reached −100 °C, and the rewarming speed slowed down due to the influence of the surrounding environment.

The temperature transformation of the side and bottom of the sample was basically consistent (such as TC4, TC5, and TC6 in Figure 1). The impact time of liquid nitrogen was basically consistent compared with the room temperature (TC7 in Figure 1). After about 3 min, the temperature decreased linearly to 0 °C due to the extremely low liquid nitrogen temperature (−196 °C) and then slowly returned to room temperature. During the experiment, the thermocouples were placed outside the wellbore wall (TC1 in Figure 1) to consider the influence of the Lyton Frost effect. The thermocouples were also placed at the liquid nitrogen impact orifice (TC2 in Figure 1). Distinctions could be observed between the two in terms of temperature changes. In the early stage of the liquid nitrogen release, the temperature change of the wellbore surface (TC1 in Figure 1) was slightly slower than

that of the liquid nitrogen impact hole (TC2 in Figure 1). This further indicated that the Lyton Frost effect slowed down the heat transfer of the specimen, thereby inhibiting the propagation of the cracks. A method for reducing the Lyton Frost effect must be established to increase the recovery rate of shale gas and improve the fracture effect of the samples.

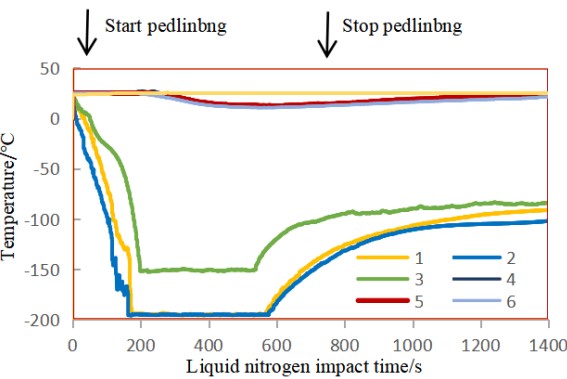

**Figure 5.** Locations of the thermocouple tips and temperature evolutions during the cryogenic thermal shock experiments (sample 1).

Figure 6 shows Cha et al.'s [16] unconfined low-pressure (about 0.01 MPa) liquid nitrogen test of the tight sandstone samples and the measured sample temperature change diagram. The temperature change was similar to the temperature change of the acrylic specimen shown in Figure 5, and the initiation and propagation of the specimen crack were related to the temperature change and distribution of the internal and surface of the specimen. During the test, liquid nitrogen further flowed into the crack, making the crack expand again, which accelerated the temperature propagation. In this regard, similar transparent materials can be used instead of shale to study the fracture mode under low-temperature freeze-cracking.

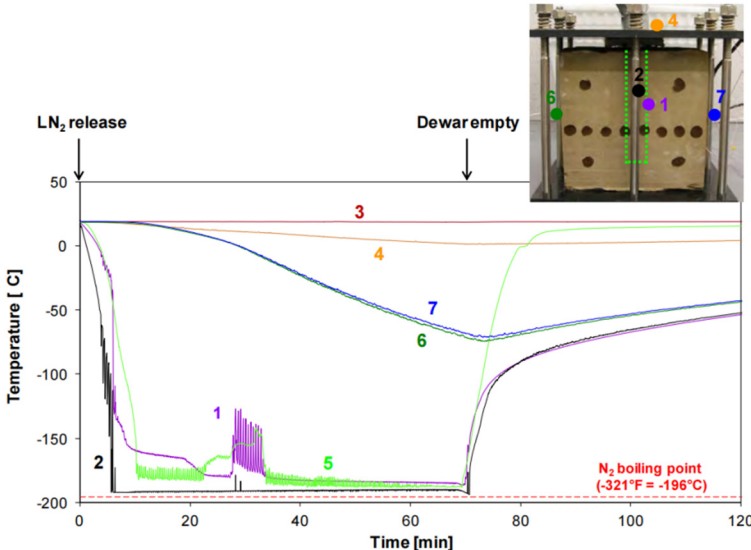

**Figure 6.** Temperature versus time during the unconfined low-pressure LN2 test. The sample picture (top right of the figure) is where the thermocouples were located on the sample (from Cha et al. [16]).

### 2.4. Crack Initiation and Propagation Process

The visualization experiment of the liquid nitrogen effect was carried out on samples 1 and 2. The settings of the two samples were consistent, except for the difference in the liquid nitrogen impact amount and impact time during the test. During the test process of specimen 1, the liquid nitrogen shocked for 9 min and consumed 0.55 kg of liquid nitrogen

(about 0.68 L), and the failure rate (the ratio of crack propagation volume to total volume) was 0.084%. Meanwhile, in specimen 2, liquid nitrogen shocked for 6 min and consumed 1.47 kg of liquid nitrogen, and the failure rate was 0.375%. The reason for this difference is that liquid nitrogen is easily boiled at room temperature. Accordingly, liquid nitrogen mainly exists in the form of gas in the first few minutes of the experiment and then mainly occurs in the form of a gas–liquid mixture. Although the impact time of sample 2 was short, the liquid nitrogen in the borehole appeared more rapidly. Then, the temperature of sample 2 immediately decreased, which resulted in a higher thermal gradient and caused higher thermal stress and failure rates.

Figure 7 shows the crack morphology and tensile stress diagram, where (a) and (b) are samples 1 and 2, respectively. Two different crack morphologies were observed: the horizontal plane crack and the vertical tensile crack. The two different crack growth modes were the horizontal plane radial propagation caused by longitudinal thermal shrinkage and the vertical propagation caused by circumferential shrinkage.

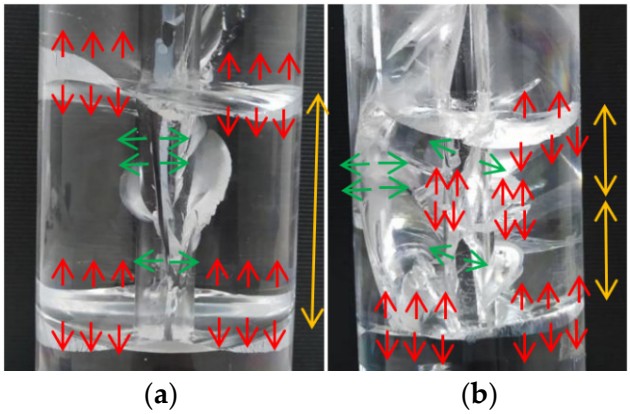

(a) (b)

**Figure 7.** Crack morphology and driving thermal tensile stresses (yellow arrows are crack spacing, green arrows indicate the circumferential shrinkage of the vertical circumferential cracks, and red arrows indicate the longitudinal shrinkage of the horizontal radial plane cracks). (**a**) specimen 1; (**b**) specimen 2.

The crack in the vertical direction originated or formed between the two horizontal cracks and bridged due to circumferential thermal shrinkage. The crack's strength was smaller compared with the crack in the horizontal direction. A certain interval length existed between the horizontal cracks because a group of cracks could not be formed within a certain length due to the limited thermal shrinkage. This kind of spacing crack also appears in other phenomena. The typical examples are the equal spacing crack of the layered rock after a long geological action, the equal spacing crack of the pavement structure under the action of temperature, ceramic cracks, rock frost heave, and ice splitting phenomena. In some rocks under certain temperature conditions, thermal shrinkage causes equidistant cracks, and the crack length and spacing also follow the law of power scaling. This characteristic can be preliminarily observed in sample 2.

Figure 8 shows the crack propagation process diagram of sample 1. During the test, the crack germination and development were accompanied by clear fracture sounds. When the tensile stress generated in the borehole reached a certain threshold, the crack rapidly expanded in a short time (the first horizontal crack at the bottom expanded for only 0.1 s, and the second horizontal crack expanded for only 0.01 s). The propagation of low-temperature cracks was sudden due to the brittleness of the specimen at low temperature and the continuous accumulation and release of the tensile stress in the borehole at the crack tip. During the two tests, a major horizontal crack first appeared at the bottom of the borehole. This phenomenon mainly occurs because, during the injection process of liquid nitrogen, because the injection point is very close to the bottom of the wellbore and is affected by gravity, it first gathers at the bottom of the wellbore, resulting in the rapid

decline in the temperature at the bottom of the wellbore. Thus, the thermal shrinkage at the bottom of the wellbore is more evident. The steel casing and epoxy resin affected the heat transfer, liquid nitrogen flow, stress distribution, and fracture distribution, resulting in the second major horizontal crack in the borehole center area. Moreover, vertical cracks were generated with the initiation and propagation of the horizontal cracks, and a bridge was erected between the two horizontal cracks. The liquid nitrogen continuously flowed into it, and the crack finally presented a complex fracture morphology under the interaction of longitudinal and circumferential thermal shrinkage.

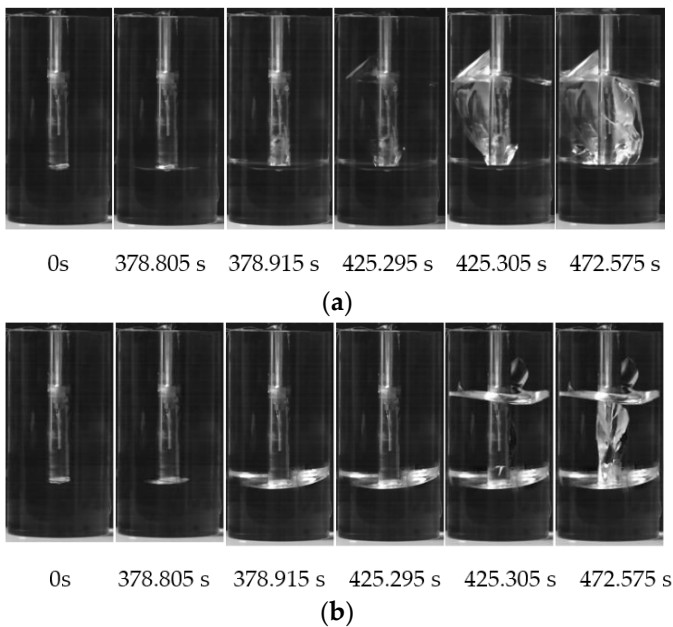

**Figure 8.** Crack propagation change during the cryogenic thermal shock experiments (sample 1). (**a**) Front; (**b**) sides.

### 3. Feasibility Study on Low-Temperature Frozen Shale Using a Numerical Simulation Method

Numerical simulations are carried out using RFPA, a software that not only takes into account the non-homogeneity of a rock mass at micro and macro levels, but also captures the key behavior of rock deformation and damage [27]. Taking the numerical calculation model of the water–ice single crack of Liu et al. [28] as an example, the thermal coupling property of RFPA2D-Thermal was used to analyze the frost heaving deformation and fracture process.

The model was 0.6 m (length) × 0.6 m (with), the cross-section length of the crack was 0.04 m, and the center width of the short axis of the crack was 0.004 m, which was a plane strain state and ensured that the model size was 15 times the crack length. When quadrilateral elements were used, the size was refined to 0.001 m, and the compression–tension ratio was 10:1. The initial temperature of the rock mass was 0 °C, and the AB edge on the crack surface was cooled to a temperature of −20 °C. The three edges were adiabatically treated, AB, with CD constraint Y direction displacement and BC constraint X direction displacement. The specific boundary conditions are shown in Figure 9. The thermodynamic parameters are shown in Table 2.

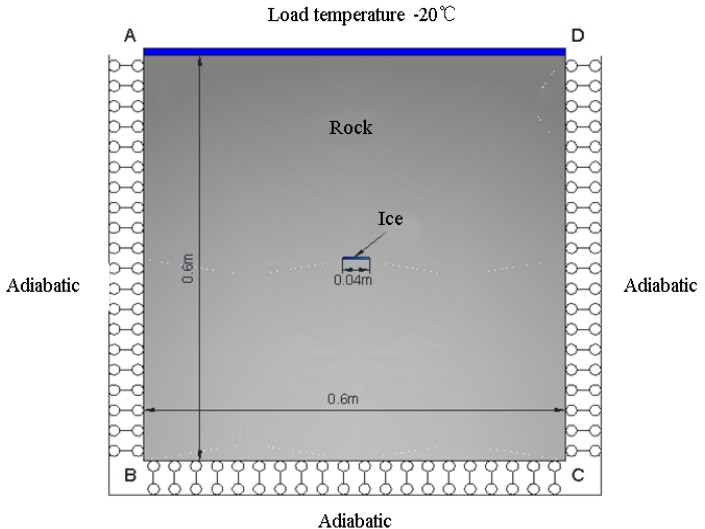

**Figure 9.** Dimensional drawing of the frost heave model.

**Table 2.** Mechanical parameters of the rock and ice.

| Parameter Name | Medium | |
| --- | --- | --- |
| | Rock | Ice |
| Compressive strength (MPa) | 100 | 20 |
| Uniformity of compressive strength | 2 | 30 |
| Poisson ratio | 0.37 | 0.35 |
| Elastic modulus (MPa) | 10,000 | 600 |
| Homogeneity of elastic modulus | 2 | 20 |
| Density (kg·m$^{-3}$) | 2410 | 917 |
| Angle of internal friction (°) | 35 | 20 |
| Coefficient of thermal conductivity (W·m$^{-1}$·°C$^{-1}$) | 2.67 | 2.2 |
| Thermal capacity (J·m$^{-3}$·°C$^{-1}$ × 10$^6$) | 1.9 | 1.7 |
| Thermal expansion coefficient (°C$^{-1}$ × 10$^{-5}$) | 0.3 | - |

Considering the effect of water migration during frost heaving, the equivalent thermal expansion coefficient method used for numerical analysis and the equivalent thermal expansion coefficient were introduced $\alpha$ [28]:

$$(1 - \alpha \Delta T)^3 = \left(1 + \beta u^T\right)(1 - \xi), \tag{1}$$

where $\beta$ is the phase change volume expansion coefficient of the water–ice under free state (0.09); the crack water begins to phase change from 0 °C, that is, the phase change point $T_f = 0$ °C; the freezing temperature is $-20$ °C, so $\Delta T = -20$ °C; $u^T = 1$; and $\xi$ represents the proportional relationship between the water transfer flux and the original volumetric water. The curve of the equivalent thermal expansion coefficient with the water transfer flux ratio was obtained by substituting Formula (1) into the numerical value, as shown in Figure 10.

Figure 11 is the minimum principal stress cloud near the crack when the water transfer flux ratio $\xi$ is 0.06. In RFPA2D-Thermal, the compressive stress was set to be positive and the tensile stress was set to be negative. An obvious tension concentration could be observed at the crack tip, and the maximum tensile stress even reached 9.84 MPa, which was less than the tensile strength of 10 MPa. The local unit failure was only caused by the heterogeneity of the rock material. The ice on the crack wall and inside the crack was mainly compressed. Given that the crack shape was non-curved and slightly changed in the numerical calculation, the stress distribution in the crack ice was not uniform. However, the stress value was in a constant state, and the overall difference was small. The tensile stress at the crack tip$\theta = 0$ continued to increase with the decrease in the freezing temperature, and

the frost heaving force caused local damage at the weak cementation of the rock mineral particles, thereby inducing a crack rupture. This condition is also one of the mechanisms by which liquid nitrogen causes the low-temperature cracking of rocks.

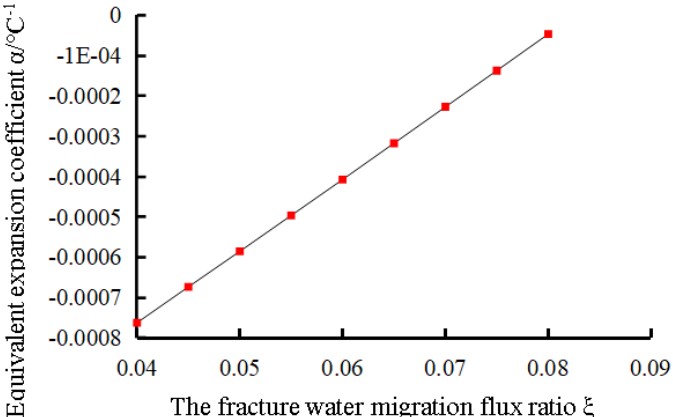

**Figure 10.** Variation of the equivalent thermal expansion coefficient with water transport flux.

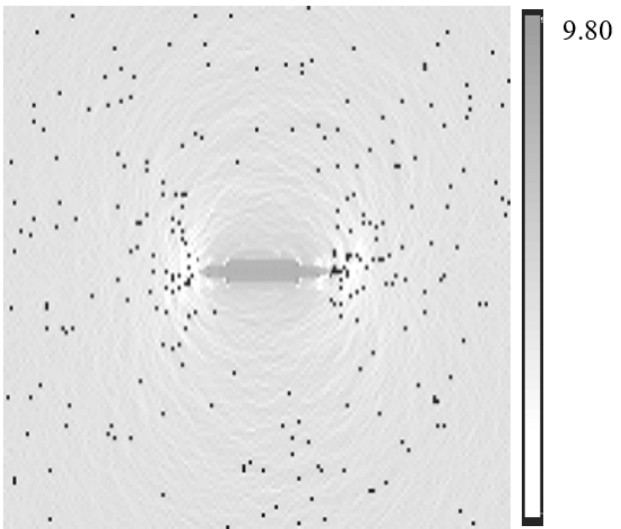

**Figure 11.** Minimum principal stress of the locally enlarged cloud image at the time of $\xi = 0.06$ (unit: MPa).

The minimum principal stress at the crack ice center was calibrated as the frost heaving force. According to the solution model deduced by Liu et al. [28], the theoretical analytical solution of the crack tip stress field under frost heaving force was compared with the numerical simulation results in this work. Figures 12 and 13 demonstrate that the numerical solution obtained by RFPA2D-Thermal is in good agreement with the analytical solution, and the change trend is also in good agreement. Hence, RFPA2D-Thermal can be used to simulate the process of freezing pore water in shale by low-temperature fluid and frost heaving damage, and it can accurately predict the crack initiation.

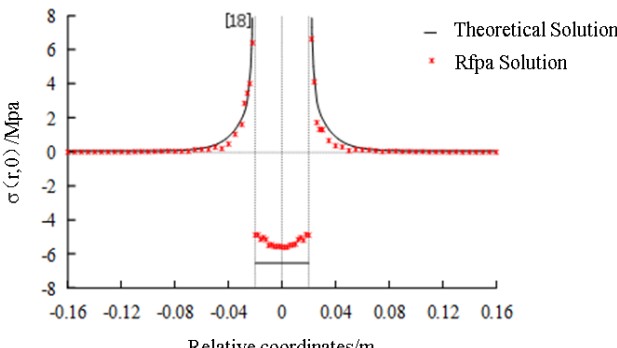

**Figure 12.** Comparative analysis of the minimum principal stress numerical and analytical solutions at $\xi = 0.06$.

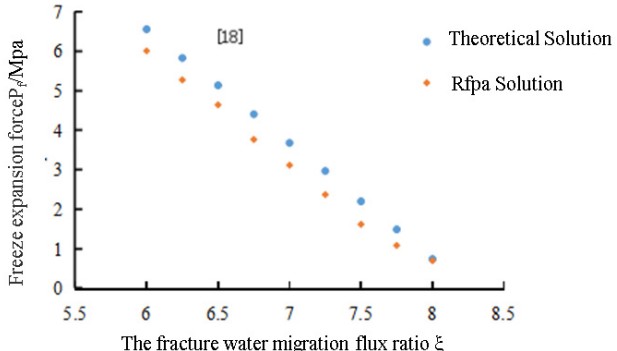

**Figure 13.** Comparison between the numerical and analytical solutions of frost heave pressure.

## 4. Numerical Simulation of Liquid Nitrogen Low-Temperature Fracturing

### 4.1. Numerical Model and Boundary Conditions of the Frost Heave

The last section shows that low temperatures can cause the pore water inside rocks to undergo a phase transition and solidify into ice. In this process, the volume expansion of water is about 9% [29]. Research shows that the force generated by the transformation of the pore water phase into ice exceeds 65 MPa [30], which is sufficient to cause pore damage. The liquid nitrogen temperature required for this experiment was much lower (about −195.8 °C) than the −20 °C temperature load simulated in the previous section. When the rock is frozen by liquid nitrogen, the freezing rate inside the rock is faster, the water transfer flux is smaller, and the frost heaving force is larger, which causes the easy breakage of the rocks.

In addition to liquid nitrogen freezing damage, liquid nitrogen cold shock is also one of the main mechanisms of liquid nitrogen low-temperature fracturing. Under reservoir conditions, the heat of the rock near the contact boundary dissipates to the low-temperature fluid under the action of cold shock and quickly cools and contracts when the low-temperature fluid makes contact with the reservoir rock. Meanwhile, cracks perpendicular to the contact boundary will be formed when the tensile stress exceeds the tensile strength of the rock [16] (Figure 14).

Given the influence of liquid nitrogen characteristics, flow velocity, and rock properties, a certain time process is necessary to reduce the temperature of the contact boundary to the same temperature as liquid nitrogen, and the temperature cracks are mainly concentrated in a certain range from the contact boundary. Most current studies directly assume that the contact boundary temperature is the liquid nitrogen temperature, which is inconsistent with the actual situation. Consequently, this work sets the contact boundary as the third boundary condition: a temperature difference exists between the rock and the liquid nitrogen at the contact boundary. The contact boundary to a lower temperature (−160–170 °C) not only needs to consider the time factor but also sets the heat transfer coefficient. In this

section, the effect of the heat transfer coefficient on the temperature and stress field during liquid nitrogen cold shock is studied by taking the liquid nitrogen freezing test under the real triaxial confining pressure used by Cha et al. [16] as an example. The required heat transfer coefficient is calculated according to the cracking effect, which provides basic data for the numerical simulation experiment considering in situ stress in the next section.

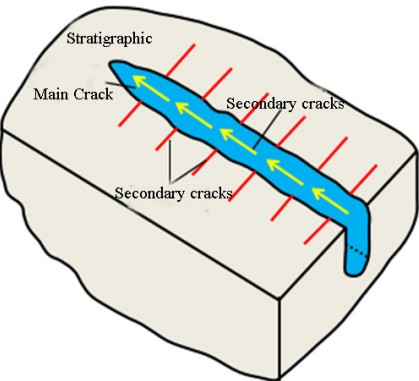

**Figure 14.** Sketch of the application of cryogenic fracturing for reservoir stimulation.

In the field test, Cha et al. [17] injected liquid nitrogen into a hole with a center diameter of 1 inch in the shale cube (8 in × 8 in × 8 in), and cracks were generated in the boundary of the hole to the rock interior under cold shock for nearly 40 min. Considering that the hole depth was much larger than its diameter, the specimen can be regarded as the plane strain state and simplified as the circular hole square plate model. The size of the model is shown in Figure 15. Quadrilateral elements were used, the number of elements was 609 × 609, and the initial temperature of the model was 20 °C. The values of the elastic modulus and Poisson's ratio were: $E_s = 49.3$ GPa and $V_s = 0.268$. Furthermore, the outer boundary of the model was not constrained, the in situ stress $\sigma = 0$, and the other parameters were the same as those in Table 2.

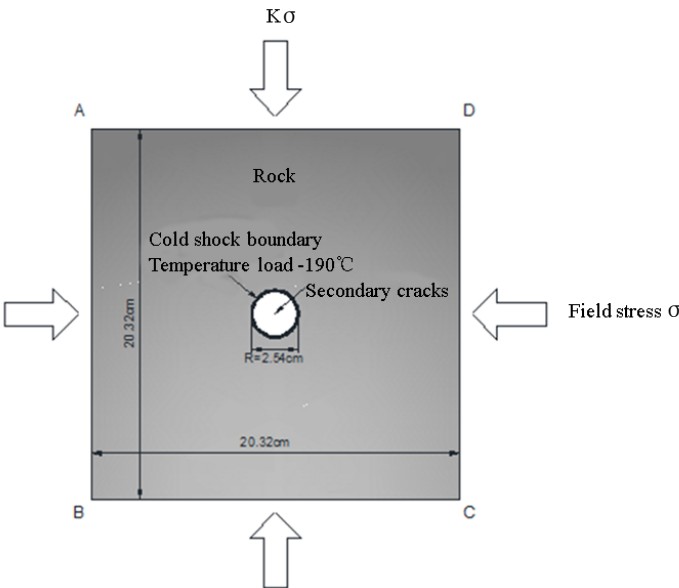

**Figure 15.** Geometry of the numerical model, where $\sigma$ is the in situ stress and $k$ is the lateral stress coefficient.

### 4.2. Effect of the Heat Transfer Coefficient on the Temperature and Stress Fields

When liquid nitrogen impacted wellbore for 40 min, the temperature and thermal stress along the wellbore radial direction evolved over time under different heat transfer coefficients ($H = 100$ W·m$^{-2}$·K$^{-1}$ and $H = 1000$ W·m$^{-2}$·K$^{-1}$), as shown in Figure 16. The temperature of

the contact boundary continuously decreased with the increase in the cold shock time, and the temperature range constantly expanded. For example, when H = 100 W·m$^{-2}$·K$^{-1}$, the contact boundary temperature decreased by nearly 17 °C in only 1 s, and the range of the radius about 18 mm from the wellbore center was affected (5 mm from the contact boundary). The temperature drop rate decreased from 1.9 °C/s to 0.25 °C/s in only >10 s and gradually slowed down. When the duration of the liquid nitrogen impact increased, the temperature range affected the whole rock, the surface temperature of the sample began to decrease, and the original large temperature gradient gradually decreased.

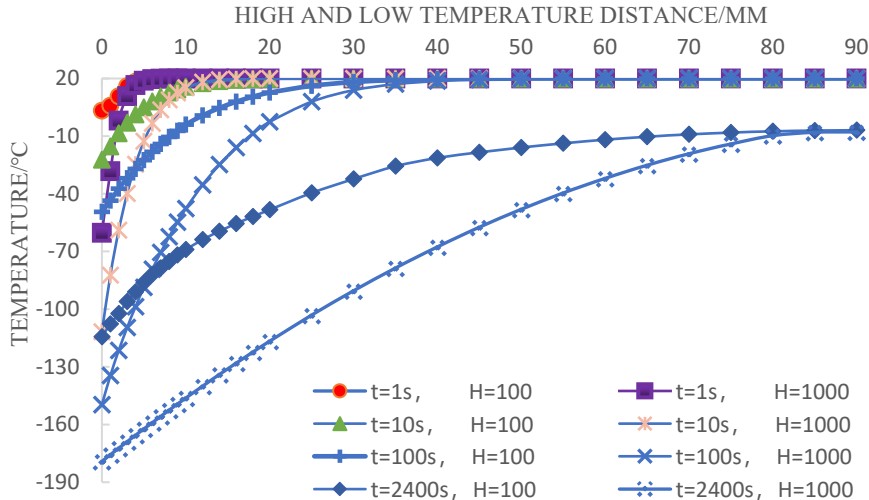

**Figure 16.** Evolution of the heat transfer coefficient of the temperature distributions on the boundary at different times.

Under the same conditions, when the liquid nitrogen was impacted for a certain time, the heat transfer coefficient was larger, the temperature drop was faster, and the temperature gradient generated by the contact boundary was greater. For example, when the liquid nitrogen and cold boundary contact time t = 1 s, the heat transfer coefficient H = 100 W·m$^{-2}$·K$^{-1}$ and H = 1000 W·m$^{-2}$·K$^{-1}$, the temperature variation range was 5 mm. However, the temperature differences between the contact interface and the sample surface were 16.78 °C and 80.27 °C, respectively, and the difference was nearly five times. After 100 s, the temperature difference of the sample with a heat transfer coefficient of H = 100 W·m$^{-2}$·K$^{-1}$ reached nearly 70 °C, which was much later than that with H = 1 000 W·m$^{-2}$·K$^{-1}$. The thermal stress of the circular hole square plate model was responsible for the thermal gradient. Specifically, a higher heat transfer coefficient made the temperature drop faster in the process of low-temperature fracturing, resulting in higher thermal gradient and stress levels. The tensile stress produced in the test reached 20 MPa, which is higher than the tensile strength of most rocks and enough to cause a low-temperature rupture.

Figures 17 and 18 further analyze the evolution process of the temperature and minimum principal stress with time spacing and the length of each crack scale according to the power law. Each crack kept the law of equal spacing and equal length, and its length was approximately twice that of the previous crack. The directions of the temperature gradients at both ends of the crack were also deflected. Meanwhile, Figure 17 (especially when t = 100 s) shows that the crack had a blocking and disturbing effect on the temperature field, the temperature conduction during crack initiation, and the propagation in the circular hole square model. Each row represents t = 0.2 s, T = 2 s, T = 20 s, and T = 100 s in the liquid nitrogen impact process. When t = 0.2 s, the temperature near the borehole significantly decreased. Figure 18c shows the low compressive stress inside the rock (less than 0.1 MPa) and the tensile stress around the borehole. The tensile stress caused by temperature change results in the formation of many microcracks below 1 mm at the contact boundary for the first time. The tensile stress began to increase, and its range continued to expand with the further increase in temperature gradient

and temperature range. From the minimum principal stress field of t = 2 s, the tip of all initiation cracks concentrated large tensile stress and connected to each other to form a tensile stress ring, driving the initiation cracks to further internally expand. In the process from t = 2 s to T = 20 s, the contact boundary temperature further decreased to $-162\ ^{\circ}$C, and the temperature range spread from 5 mm to 12 mm. The low-temperature cracks generated in the liquid nitrogen impact process radially propagated along the center of the borehole with the temperature range. However, the maximum temperature gradient gradually decreased from the contact boundary to the inner part of the rock due to the low thermal conductivity of the rock. The originally initiated microcracks (about 5 mm in length) gradually evolved into a grading phenomenon within 100 s, as shown in Figure 18c. After 20 s, three of the eight second-order cracks (about 12 mm in length) stopped expanding. The other five cracks were third-order cracks (about 25 mm in length after 100 s). The crack distance between the cracks was uneven, and the temperature field was not smooth due to the influence of the main crack propagation.

In Figures 17 and 18a–c, the heat transfer coefficients are H = 100 W·m$^{-2}$·K$^{-1}$, H = 500 W·m$^{-2}$·K$^{-1}$, and H = 1000 W·m$^{-2}$·K$^{-1}$, respectively. From the above conclusions, the higher the heat transfer coefficient, the faster the temperature drop rate of the contact boundary within the same temperature range, and the greater the temperature gradient generated by the cooling. A larger temperature gradient means a larger thermal stress. Hence, rock cracks are easier to crack, and more cracks are formed.

The larger the heat transfer coefficient, the faster the peak transfer of the minimum principal stress, the larger the tensile stress range, the larger the crack propagation range, and the deeper the crack depth (t = 100 s). The three heat transfer coefficients from low to high were 18, 25, and 33 mm. Therefore, to improve the heat transfer coefficient in the process of liquid nitrogen low-temperature fracturing, the damage speed was faster, and the damage range was larger for the exploitation of low-permeability shale gas.

*4.3. Effect of the Heat Transfer Coefficient on the Crack Propagation*

The monitoring of acoustic emissions in rock can reflect the changes in internal structure, help us to understand the time, location, and strength of microfractures, and further reveal the basic laws of rock crack initiation and propagation. Accordingly, the process is divided into various stages throughout the process of liquid nitrogen cold shock damage to the circular orifice plate model under different heat transfer coefficients, according to the changes in acoustic emission quantity and tensile stress: (1) the linear elastic deformation stage; (2) the microcrack initiation to propagation stage; (3) the instability fragmentation stage, or the stop expansion stage. Figure 11 shows the temporal variations in the number and cumulative number of acoustic emissions during liquid nitrogen cold shock, which is not described in the figure due to the rapidness of the first stage. In the second stage, the graded cracks around the wellbore are formed within 100 s. Thus, the second stage is analyzed based on 100 s.

Figure 19 shows that the cumulative number of acoustic emissions increases with the cooling time, and the growth rate rapidly increases first and then fluctuates. Microcracks appeared in the rock almost within 0.2 s, which is very rapid. A large number of acoustic emission events mainly occur in the first 5 s. The reason is that the contact boundary is directly subjected to cold shock at this stage. The rapid increase in temperature gradient results in the instantaneous increase in tensile stress at the contact boundary, resulting in nearly a dozen microcracks rapidly initiating and propagating along the radial direction of the borehole (microcrack propagation period). After 5 s, some microcracks stopped propagating and gradually evolved into graded cracks. However, the number of AE events showed a certain fluctuation decrease at this stage, indicating that the crack growth rate slowly decreased over time. The higher number of acoustic emissions at a certain time was mainly due to the uneven distribution of the material properties caused by the heterogeneity of the rock materials, and more point damage occurred around the crack tip.

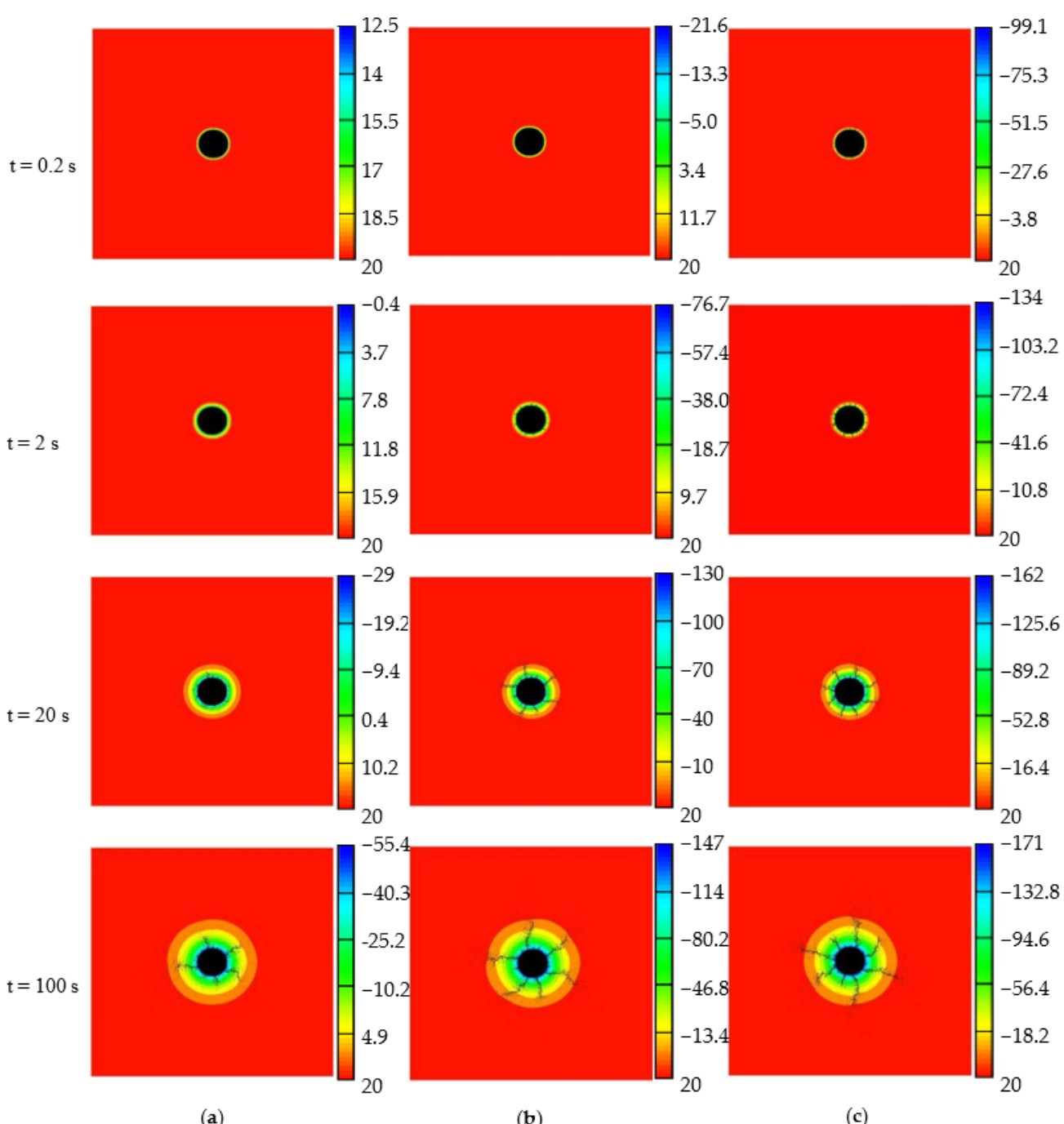

**Figure 17.** Temperature distribution and crack propagation in the circular hole plate models with different heat exchange coefficients (unit: °C). (**a**) H = 100 W · m$^{-2}$ · K$^{-1}$; (**b**) H = 500 W · m$^{-2}$·K$^{-1}$; (**c**) H = 1000 W · m$^{-2}$ · K$^{-1}$.

Rocks with low heat transfer coefficients (H = 100 W·m$^{-2}$·K$^{-1}$) showed obvious hysteresis during a liquid nitrogen cold shock. Figure 17 demonstrates that the failure elements appeared only after 1 s. The number of these random and disordered point-like failure elements continuously increased after 5 s, turning to microcracks. This was slower than that of rock with a high heat transfer coefficient at 0.2 s. Hence, the number of acoustic emissions in the first few seconds of rock with a low heat transfer coefficient was smaller than that of the rock with a high heat transfer coefficient, showing a fluctuating upward trend. At 20 s, some microcracks stopped propagating and gradually evolved into graded

cracks. This phenomenon further shows that the larger the heat transfer coefficient, the faster the temperature gradient changes, and the faster the rock failure rate.

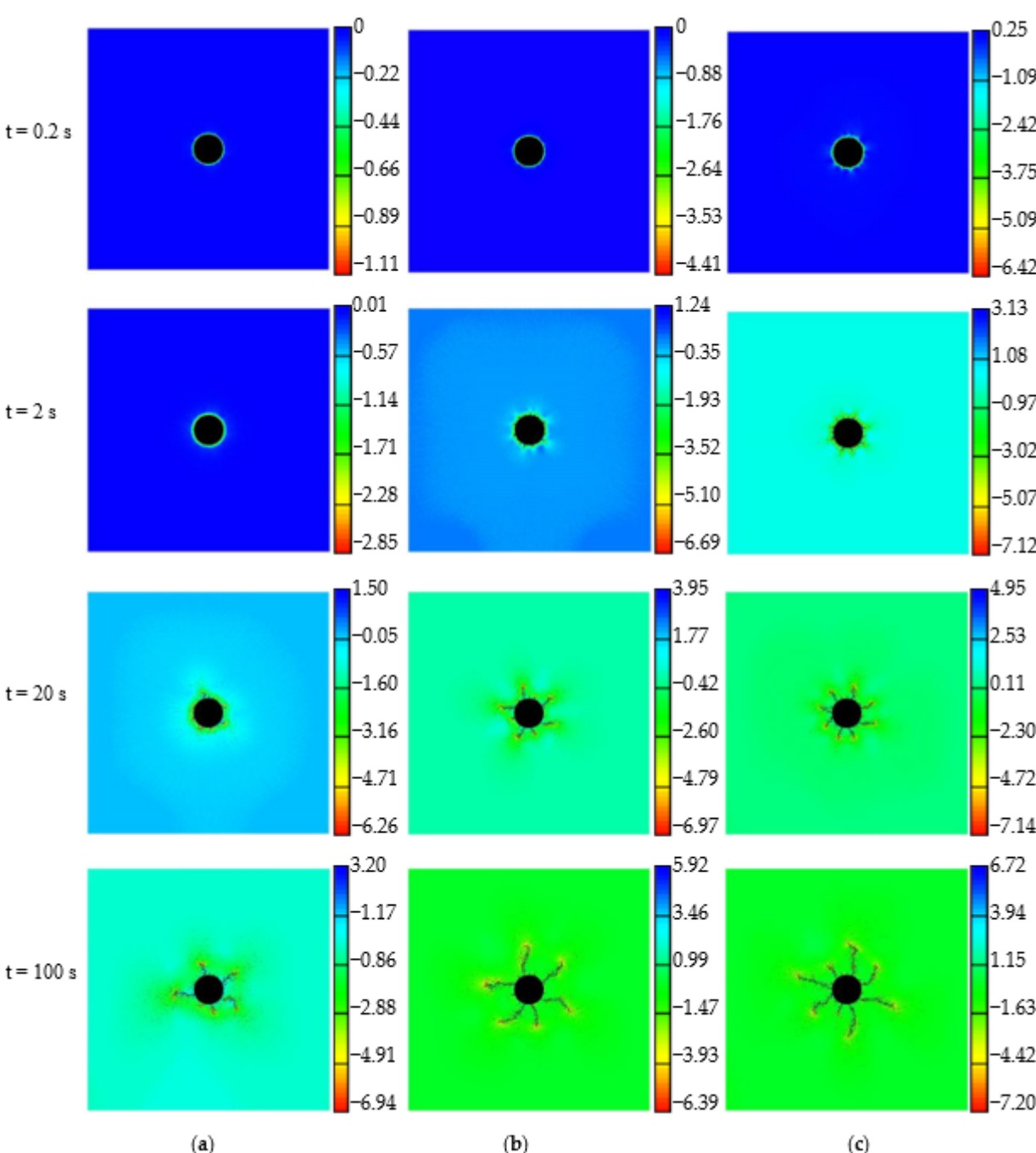

**Figure 18.** Minimum principal stress and crack propagation process in the circular hole plate models with different heat exchange coefficients (unit: MPa). (**a**) H = 100 W · m$^{-2}$ · K$^{-1}$; (**b**) H = 500 W · m$^{-2}$ · K$^{-1}$; (**c**) H = 1000 W · m$^{-2}$ · K$^{-1}$.

After a long crack propagation period, the rock enters the third stage, namely, the unstable fracture stage, or the stop propagation stage. Given the high elastic modulus and strength of rock, the above model needs at least 1800 s to achieve this stage. Table 3 illustrates the statistical table of the failure rate of rocks with various heat transfer coefficients. The failure rate of the rocks with high heat transfer coefficients (H = 1000 W·m$^{-2}$·K$^{-1}$) within 100 s was 0.84% (the proportion of the failure units in the total unit), accounting for

27.23% of the final failure rate, and indicating that the propagation range of cracks from t = 1 s to t = 100 s was much larger than that from t = 100 s to t = 2340 s. After 100 s, the temperature gradient continued to decrease with time, and the propagation rate of the cracks gradually slowed down.

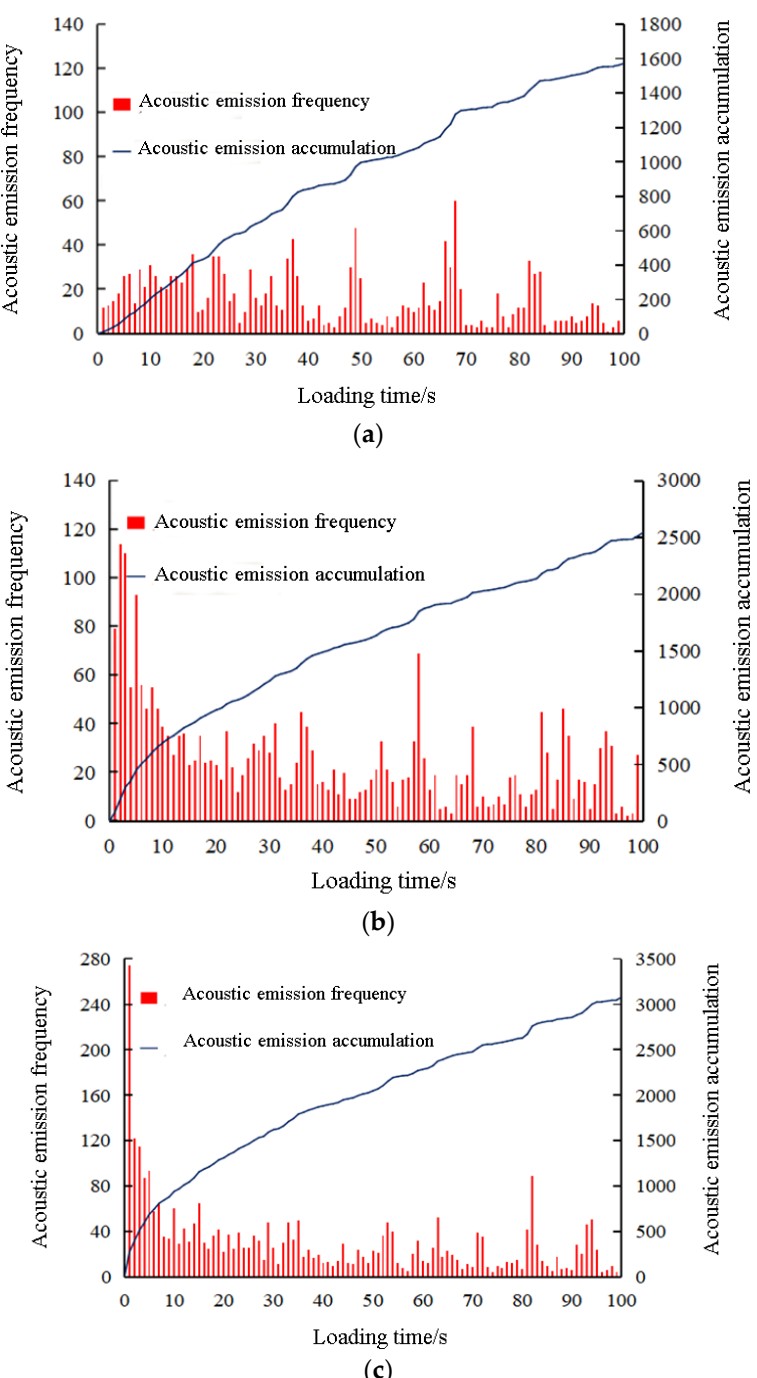

**Figure 19.** Acoustic emission changes under the condition of thermal shock by soaking the specimen in liquid. (**a**) Heat transfer coefficient H = 100; (**b**) heat transfer coefficient H = 500; (**c**) heat transfer coefficient H = 1000.

**Table 3.** Failure rate of rocks with different heat exchange coefficients.

| Coefficient of Heat Transfer | H = 100 W·m$^{-2}$·K$^{-1}$ | H = 500 W·m$^{-2}$·K$^{-1}$ | H = 1000 W·m$^{-2}$·K$^{-1}$ |
|---|---|---|---|
| Stop extension time/s | 1800 s | 2000 s | 2340 s |
| Damage rate at 100 s/% | 0.430 | 0.695 | 0.841 |
| Final destruction rate/% | 2.096 | 2.677 | 3.089 |
| 100 s failure rate ratio/% | 20.51 | 25.97 | 27.23 |

*4.4. Discussion of Engineering Measures to the Improve Heat Transfer Coefficient*

In summary, a high heat transfer coefficient has an important effect on the temperature and stress distribution in a rock. The greater the heat transfer coefficient is, the greater the temperature gradient generated by the liquid nitrogen cold shock and the higher the corresponding temperature stress will be. However, the insulating gas layer will be formed at the contact interface when liquid nitrogen enters the rock due to the influence of the Lyton Frost effect, thereby hindering the decrease in surface temperature, which is also the reason for the low heat transfer coefficient of liquid nitrogen. When it impacts low-permeability gas reservoirs (such as shale gas reservoirs), it may not produce a large number of cracks, as expected due to the small stress variation. Accordingly, physical tests may require longer liquid nitrogen freezing times, multiple freezing cycles, or a higher liquid nitrogen impact pressure (Figure 20) to observe obvious cracks on the sample surface. The influence of the Lyton Frost effect must be eliminated, and the heat transfer coefficient must be improved to increase the degree of rock fracturing. Common engineering measures include increasing the liquid nitrogen flow rate and liquid nitrogen pressure. The Lyton Frost effect occurs at relatively high temperatures with the increase in pressure. Cha et al. [16] showed that the temperature drop on the sample surface will increase with the increase in liquid nitrogen flow rate. Nitrogen fixation and fracturing proppant as solid suspension in liquid nitrogen can also be used. Such a mechanism helps in reducing the Lyton Frost effect, and proppant particles embedded in shale rock plates affect the conductivity of supporting fractures [31], which is conducive to maintaining the opening of fractures.

Although the liquid nitrogen transformation of shale gas reservoirs has good prospects, some oil and gas exploitations used as fracturing fluids have shown good results in the expansion of main cracks and the manufacturing of secondary cracks. However, many technical barriers need to be overcome because the current research is still in the stage of theoretical research and process design. Moreover, the transformation of fracturing fluid from water to liquid nitrogen involves commercial costs, even though this process is more economical and effective for the development of shale gas, and rigorous trade-offs and comparisons are still needed.

Front end of cracks generated by

low temperature

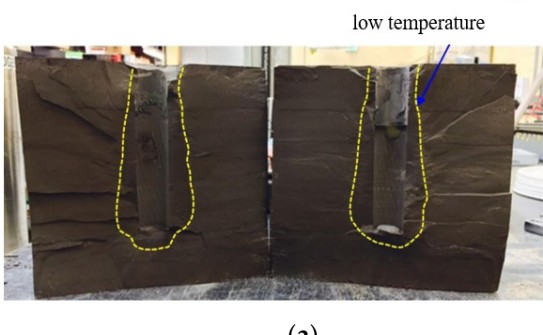

(**a**)

**Figure 20.** *Cont.*

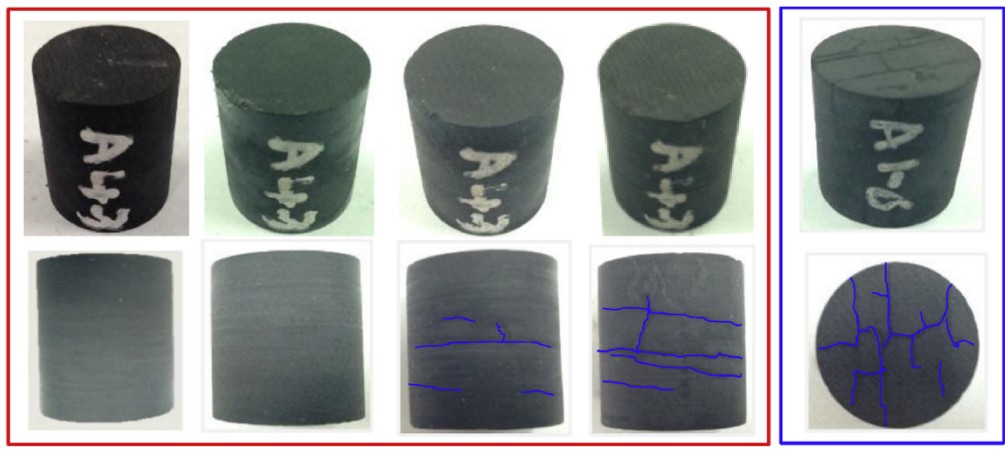

(a₁)one freeze-thaw cycle  (a₂)two freeze-thaw cycles  (a₃)three freeze-thaw cycles  (a₄)four freeze-thaw cycles  (b₁)four freeze-thaw cycles

Vertical bedding direction

Parallel bedding direction

**(b)**

**Figure 20.** Failure patterns of shale under the condition of thermal shock by soaking the specimen in liquid nitrogen. (**a**) Test results of Cha [17]; (**b**) test results of Han [32].

## 5. Effect of In Situ Stress on Crack Propagation

Given the varying shale physical properties and in situ stress conditions, different fracture forms, including single fracture, complex fractures, and network fractures, may be formed after fracturing [33]. The in situ stress state is an important factor that affects crack initiation and propagation. Accordingly, understanding the influence of in situ stress on crack propagation and predicting and controlling the crack direction to ensure that it can communicate with natural cracks and form complex network cracks are the keys to improving the volume of fracturing. Based on the circular hole model of the horizontal well section, the in situ stress σ (x direction) is set as 6.89 MPa (1000 psi) to study the influence of temperature stress distribution and crack propagation under the liquid nitrogen impact of the circular hole model with different lateral stress coefficients k (k = 0.25, 0.5, 1, 2).

The experimental results reported by Cha et al. [16] show that the surface temperature of the borehole decreased to a lower temperature (−160–170 °C) after 10 min of a low-temperature and low-pressure injection of liquid nitrogen. After 40 min of impact, the surface temperature of the sample decreased from 20 °C to −20 °C, which is nearly 40 °C. In combination with the above discussion results and Figure 8, the heat transfer coefficient was preliminarily selected as 500 W·m⁻²·K⁻¹. The other parameters were the same as those of the model experiment in Section 2.

The propagation depths of the cracks generated at low temperature in the direction of larger principal stress were longer, as shown in Figure 21. For example, when k = 0.25, that is, the constraint stress kσ in the y direction was 1.7225 MPa, this was much smaller than the constraint stress in the x direction. The maximum main crack propagated along the x direction. When t = 100 s, the propagation length of the main crack could reach 35 mm, while the crack in the y direction was almost invisible to the naked eye. When k = 2, the constraint stress kσ in the y direction was 13.78 MPa, which was much larger than that in the x direction. The maximum main crack propagated along the y direction. When t = 100 s, the propagation length of main crack was about 9 mm, while the propagation length of the crack in the x direction was only 1 mm. This phenomenon mainly occurs because the reaction to the applied temperature load is small when the rock is restricted. Meanwhile, the tensile stress in the restricted direction is small when the liquid nitrogen is cold-shocked. When k < 1, the tensile stress in the y direction of the borehole was less than that in the x direction, as shown in Figure 13. Cracks occurred when the value was greater than the tensile strength of the rock. The cracks in the y direction stopped propagating with the development of time. Meanwhile, the cracks in the x direction continued to develop. However, in the case of k < 1, the constraint stress in the direction of k = 0.25 was smaller

than that in the direction of k = 0.5 y, and the restriction was smaller. Although the number of cracks was less, the propagation depth of the main crack was deeper. In this case, the crack was not uniformly distributed, and its length was not similar or graded compared with the crack generated when k = 1.

In the actual engineering operation, the above model corresponded to the horizontal well area. Under the condition of a certain horizontal geostress, the propagation depth of the crack generated by low temperature along the larger stress direction can be used to change the vertical stress (converted k value) through ground loading and other methods to control the crack propagation direction and ensure that it can provide valuable guidance for the field application of low-temperature fracturing technology and the selection of the most effective operating factors.

After 40 min of liquid nitrogen cold impact, the length of the main crack generated under the condition of K = 2 was about 13 mm [17], similar to the length of the crack after 40 min in Figure 9a, indicating the feasibility of the numerical simulation software to simulate the low-temperature fracturing process of liquid nitrogen.

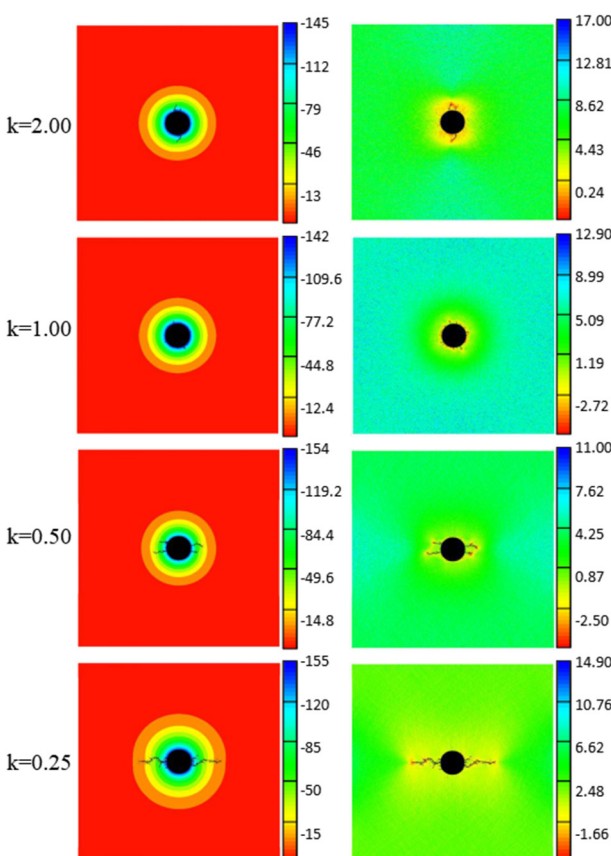

**Figure 21.** Effect of different lateral stress coefficients on temperature, minimum principal stress distribution, and crack propagation of the circular hole model under liquid nitrogen impact (unit: MPa).

## 6. Effect Mechanism of the Secondary Cracks on Liquid Nitrogen Cracking

When the fracturing effect of single liquid nitrogen is not ideal, the fracturing effect can be improved by repeated or multiple liquid nitrogen cold shock freeze–thaw cycles. The secondary cracks generated after single low-temperature liquid nitrogen fracturing further expand under cold shock, as shown in Figure 14. However, the previous models assume that liquid nitrogen will not penetrate into the cracks, and that the cracks will not interfere with the transient temperature distribution in this arrangement, without considering the influence of secondary cracks. Accordingly, one of the crack surfaces after a single liquid nitrogen cryogenic fracturing is taken as an example in this section, and the

temperature conduction process can be simplified into the model shown in Figure 22. The size of the model was 55 mm × 55 mm. The spacing of the three secondary cracks was 18 mm, and they were all sharp corner cracks with a length of 12 mm and width of 1 mm. The parameters were the same as those in Section 1 of Table 1.

　　　Figure 22 shows the temperature stress and crack propagation distribution of the crack model when t = 20 s when the low-temperature crack was subjected to liquid nitrogen cold shock again. The low-temperature crack was affected by the cold shock of liquid nitrogen. The secondary cracks generated in the middle part of the crack were interconnected, which further increased the contact area between the reservoir and the borehole. In combination with the first low-temperature fracturing crack, a dense network was formed, which improved the reservoir reconstruction volume. In particular, the low-temperature fracturing of liquid nitrogen occurred mainly in the first 100 s, and the crack propagation stopped at t = 1800 s. The 1800 s time was 30 min, which is a very small time scale for a single well shale gas development. In theory, the crack opening will constantly increase as long as the temperature of the model continues to decline, which is not related to the increase or decrease in temperature or gradient. This finding shows that cold shock has instantaneous and long-term effects on reservoir reconstruction.

　　　The shale gas recovery is more economical and effective when the aforementioned characteristics are used. In the early stage, the cracks can be propagated by a liquid nitrogen high pressure jet, and the propagation pressure can be reduced. In the middle stage, the liquid nitrogen is injected into the existing cracks again. The temperature cracking is used to stimulate the secondary cracks, increase the crack density, and form a dense crack network. The combination of instantaneous high pressure and long-term low pressure can be adjusted according to the actual site to ensure that the contact area between the reservoir and the borehole is increased as much as possible under cost control to produce energy.

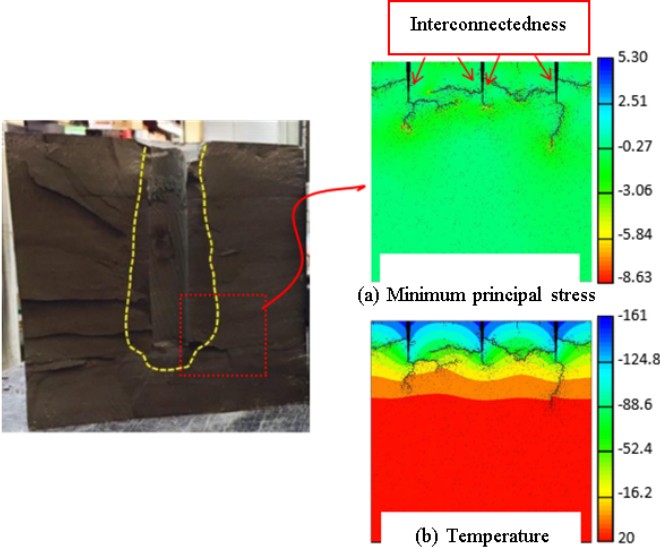

**Figure 22.** Temperature distribution and the minimum principal stress and crack propagation process under the condition of thermal shock for the second time.

## 7. Discussion

　　　According to the numerical simulation results of the homogeneous materials and the field test results, the overall crack propagation law is still very similar, even though some differences in temperature change and crack propagation roughness can be observed. For example, horizontal cracks perpendicular to the liquid nitrogen interface are generated, and the crack spacing is equal. If the influence of homogeneity and material brittleness on the crack propagation mode of low-temperature fracturing can be further obtained, then using homogeneous transparent materials to study rock low-temperature fracturing has high reliability and feasibility.

The liquid nitrogen transformation of shale gas reservoirs has good prospects, and some oil and gas exploitation methods can be used for fracturing fluids. These have shown good results in the expansion of main cracks and the manufacturing of secondary cracks. However, many technical barriers still need to be overcome because this research is still in the stage of theoretical research and process design. Moreover, the transformation of fracturing fluid from water to liquid nitrogen involves commercial cost issues, despite being economical and effective for the development of shale gas, and rigorous trade-offs and comparisons are still needed. In addition, while this paper has achieved some meaningful research on the above aspects, the influence of other factors need to be better considered in light of the many other potential variables.

## 8. Conclusions

(1)  Two different crack patterns were observed in the field test: horizontal plane cracks and vertical tensile cracks. The two different crack growth modes were horizontal plane radial propagation caused by longitudinal thermal shrinkage and vertical propagation caused by circumferential shrinkage. The cracks rapidly expanded in a short time, and a certain interval length could be observed between the horizontal cracks;

(2)  The crack propagation of the samples with different homogenization degrees was basically distributed in the direction of the vertical liquid nitrogen contact surface. When the homogenization degrees were $m = 2$ and $5$, the crack surface was rough and the microcracks were disorderly and distributed in a point-like manner around the crack tip. When $m \geq 10$, the point-like damage around the crack tip did not appear, and the crack propagation was close to the results obtained using homogeneous materials;

(3)  Using homogeneous transparent materials to study rock low-temperature fracturing has high reliability and feasibility;

(4)  RFPA2D-Thermal has good feasibility for simulating rock liquid nitrogen cracking, and it can accurately predict crack initiation;

(5)  Under the other conditions, the larger the heat transfer coefficient was, the faster the temperature drop rate of the contact boundary, and the larger the temperature gradient generated by the cooling when the liquid nitrogen was cold-shocked for a certain time. This resulted in the increase in the tensile stress value and the range of the contact boundary, thus making the rock easier to crack, and the crack initiation effect better (the number of cracks was greater, the failure rate was faster, and the failure range was larger). Thus, increasing the liquid nitrogen flow rate and liquid nitrogen pressure or using nitrogen fixation and fracturing proppant as a solid suspension in liquid nitrogen are effective measures to reduce the Lyton Frost effect and improve the heat transfer coefficient;

(6)  The lateral stress coefficient affects the propagation direction of cracks. The propagation depths of low-temperature cracks are deeper in the direction of larger principal stress, and the cracks are concentrated in the direction of larger principal stress under different lateral stress coefficients. In the field application, the vertical stress can be changed by ground stacking or loading to control the direction of crack propagation;

(7)  Cold shock has instantaneous and long-term effects on rock crack transformation. According to the field practice, the combination of instantaneous high pressure and long-term low pressure can help in the design of a more economical and effective liquid nitrogen low-temperature fracturing scheme.

**Author Contributions:** Conceptualization, C.B. and M.Z.; methodology, C.B.; software, Y.C.; data curation, M.Z.; writing-original draft preparation, C.B., M.Z. and Y.C. All authors have read and agreed to the published version of the manuscript.

**Funding:** This research received no external funding.

**Institutional Review Board Statement:** Not applicable.

**Informed Consent Statement:** Not applicable.

**Data Availability Statement:** Not applicable.

**Conflicts of Interest:** The authors declare no conflict of interest.

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
