# Peer review of "Study of Rock Crack Extension under Liquid Nitrogen Low-Temperature Fracturing"

_applsci, doi:10.3390/app12115739_

Round 1

Reviewer 1 Report

This paper is a good attempt to address the fracturing mechanism in the shale roco using liquid nitrogen.  The paper is well written and supplemented with sufficient figures to support the results and interpretation. After careful reading of the manuscript, I came to the conclusion to suggest acceptance in the present form.

Author Response

Thank you for your comments

Reviewer 2 Report

The article is devoted to the actual topic. Using the authors results of research can improve the efficiency of the development of shale gas resources. The process and results of the studies performed by the authors are well described and demonstrated in the article.

The following questions about this article:

- Information about shale gas should be added to the abstract and keywords.

- It's not clear what fig. 2-1 (105) and tab 2-1 (107) means in the text? There are no tables and figures with such numbers in the article.

- To fig. 1 should be supplemented with information about, what the colored dots and numbers in the figure mean.

- There is no link to Fig.2 in the test of the article.

- On fig. 4 there are text not by english.

- In fig. 5, the inscription near the arrow is not visible.

Reviewer 3 Report

Line 28-29, remove the keywords from the Abstract section.

Line 105-107, Fig. 2-1 and Table 2-1? Do you men Fig. 1 and Table 1?

Line 126, do you mean Fig. 2 or Fig. 3?

In Fig. 4, delete not English letters or words

Fig.5, has low quality. 

Line 200, do you mean Fig 3-7 or Fig. 7?

Reviewer 4 Report

Dear Author:

Considering the manuscript with the manuscript ID: applsci-1742203, entitled "Study of Rock Crack Extension under Liquid Nitrogen Low Temperature Fracturing", this paper is well elucidated and well referenced, nevertheless, some more discussions need to be added to improve the comprehensiveness of demonstration of the method and some needs to be removed, revised or edited. Some minor issues need to be addressed or corrected to improve the general quality and readability of this article. The paper needs to be developed in some parts. So based on my comments that comes in the following, I proposed the paper to be subjected for moderate revision.

Best Regards

General Comments: (No need to be responded by author, but to be considered during the revision) 

Is the paper new, technically correct, and relevant?

Partially yes, the paper is new and technically sounds. Results somehow does support the methodology, but needed to be more cleared by the author in case of properties the data.

Is the paper well organized?

The paper is properly organized, good literature review, suitable motivation and clear explanation on results are positive points to that.

Is the abstract concise?

Yes, but I think it needs to be rephrased after revision to add some comments about any artifacts or negative points in the method, if exist. 

Is the introduction motivating?

Yes, Introduction section is motivating. 

Are the methodology, results, and conclusions completely developed?

No, they need to be modified and developed according to the technical comments.

Are there language, mathematics, reference, or style errors? There is no mathematical, reference or style error. 

Technical Comments: 

What for a long manuscript. It was a comprehensive study that required large time to be read carefully and being reviewed. The theoretical background and also the strategy of the method has been well explained in details, and large number of experiments and diagram are presented. However, adding a flowchart or a paragraph for better explanation could result in a better understanding the whole procedure that a reader will face when he starts reading this paper. It could be placed at the end of the introduction section.

The result comparison parts are well organized and presented. The display way is good. But quantitative evaluation simulations are missed. For that section, some other quantitative evaluation need to be done, which I could not find them in the manuscript, or any explanation about them. You can add these studies during the revision or use results from those studies performed in this field or put some explanation about them.

The models are homogenous or does not have azimuthal heterogeneity. Please gives some explanation about the simplification of heterogeneity in your experiment, some explanation about the effect of various degree of heterogeneity and more importantly the effect of the azimuthal anisotropy and also the influence of the angle between the fracture propagation and the largest and smallest radius effect or orientation of the minor and major axes of the ellipse of anisotropy.

Can you model the fracture density, fracture aperture and other related parameters to the fracture after your experiment?  

Uncertainty analysis. One of the major steps in any simulation analysis and modelling study, is performing analysis of the uncertainty analysis. So, please gives some evidence of the uncertainty analysis and the studies that you have done to reduce this uncertainty.

The sensitivity analysis. After or during the uncertainty analysis, sensitivity analysis needed to be considered. In this way, you will define that which parameter has the highest influence on your results. Then, by managing the parameter with the highest negative influence, you can reduce the uncertainty in your modelling procedure. Did you consider such procedure?

To better evaluate fracture behavior, you need to fully model the behavior of fracture through different pressure regimes. Do you think that you need to perform such analysis in detail for any type of fracture and aperture size? The published manuscript under title: “Naturally fractured hydrocarbon reservoir simulation by elastic fracture modeling” published in Petroleum Science, 2017, could give you some idea about that. That publication could be cited to give the reader to come to new ideas by integrating your experiment results and modelling behavior of fractures for carbonate reservoirs.

The authors should explain what limitations did they find out about the experiments.

What software is used to perform the simulation?

In thermal hydraulic fracturing, we consider two topics: A) If thermal fluid join in the fracturing? And B) Do we need to consider the fluid overpressure? Do we need the above-mentioned issues in your experiment?

As a recommendation, since the researchers are moving towards fully automatic simulation, analysis and processing of results and data, and one of the major applications of fracturing is in petroleum reservoir for increasing the production index of the fractured reservoir, it came to my mind that the presented experiment and simulation has the ability to be expanded by one of the expert system methods. This idea might also come to the mind of any reservoir engineer who read this paper and has experience in reservoir engineering, fracture simulation and expert system analysis. So, to help to provide such idea to the readers of your paper, I propose to read and cite the publication under title: “Integration of knowledge-based seismic inversion and sedimentological investigations for heterogeneous reservoir” published in Journal of Asian Earth Science,2021.

Besides that, as another idea in application of the presented method in petroleum production, more idea came to my mind in the field of increasing the flow rate of the fluid around the production well. I can also propose to have look and cite the publication under title: “Well performance optimization for gas lift operation in a heterogeneous reservoir by fine zonation and different well type integration” published in Journal of Natural Gas Science and Engineering, 2017. Your method can bring some ideas to perform fracturing by the proposed method in heterogenous reservoir.

How to know the dilatation degree (width) of fractures? And how to know the curvature of fractures?

Best Regard

Author Response

请参阅附件
